A high-performance, hardware-based deep learning system for disease diagnosis

Siddique Ali 1 2
Iqbal Muhammad Azhar 3
Aleem Muhammad 4
Lin Jerry Chun-Wei jerrylin@ieee.org 5
1 National University of Computer and Emerging Sciences , Lahore Campus , Pakistan
2 University of Macau , Taipa , Macau
3 Lancaster University , Lancaster , United Kingdom
4 National University of Computer and Emerging Sciences , Islamabad , Pakistan
5 Western Norway University of Applied Sciences , Bergen , Norway
Sohaib Osama
Electronic publication date: 2022 Jul 19
Publication date: 2022
Volume: 8
Electronic Location ID: e1034
Received 2022 Mar 9; Accepted 2022 Jun 20
Copyright: ©2022 Siddique et al.
Copyright year: 2022
Copyright holder: Siddique et al.
License: This is an open access article distributed under the terms of the Creative Commons Attribution License, which permits unrestricted use, distribution, reproduction and adaptation in any medium and for any purpose provided that it is properly attributed. For attribution, the original author(s), title, publication source (PeerJ Computer Science) and either DOI or URL of the article must be cited.
License URL: https://creativecommons.org/licenses/by/4.0/

Keywords: Activation function, Cancer diagnosis, Deep learning, Field programmable gate array, Hardware friendly, Neural networks, Swish

Funding: The authors received no funding for this work.

==============================
Modern deep learning schemes have shown human-level performance in the area of medical science. However, the implementation of deep learning algorithms on dedicated hardware remains a challenging task because modern algorithms and neuronal activation functions are generally not hardware-friendly and require a lot of resources. Recently, researchers have come up with some hardware-friendly activation functions that can yield high throughput and high accuracy at the same time. In this context, we propose a hardware-based neural network that can predict the presence of cancer in humans with 98.23% accuracy. This is done by making use of cost-efficient, highly accurate activation functions, Sqish and LogSQNL. Due to its inherently parallel components, the system can classify a given sample in just one clock cycle, i.e., 15.75 nanoseconds. Though this system is dedicated to cancer diagnosis, it can predict the presence of many other diseases such as those of the heart. This is because the system is reconfigurable and can be programmed to classify any sample into one of two classes. The proposed hardware system requires about 983 slice registers, 2,655 slice look-up tables, and only 1.1 kilobits of on-chip memory. The system can predict about 63.5 million cancer samples in a second and can perform about 20 giga-operations per second. The proposed system is about 5–16 times cheaper and at least four times speedier than other dedicated hardware systems using neural networks for classification tasks.

Introduction

Deep learning is a subset of machine learning that does not involve much human effort and does not require handcrafting of features (Guo et al., 2019). In fact, by using deep learning techniques, machines and systems learn by themselves. It is important to note that deep learning and neural networks are not two separate ideas or techniques; any neural network that has two or more layers is considered ‘deep’. Neural networks find applications in stock market prediction, agriculture, medical sciences, document recognition, and facial recognition, among others (Awais et al., 2019; Nti, Adekoya & Weyori, 2021; Zhou et al., 2019; Guan, 2019; Chen et al., 2020; Kim et al., 2020; Lammie et al., 2019). The process of learning is usually carried out using ‘backpropagation’, a supervised learning technique in which the parameters of a neural network are adjusted according to a predefined error function. The parameters that give the lowest error at the output are selected as the optimal parameters.

It must be noted that hardware throughput is directly dependent on underlying algorithms. Therefore, efficient ANN algorithms and activation functions need to be devised if real-time neural processing is required. Sometimes, accuracy has to be sacrificed to support low-delay classification at low costs. The required level of accuracy, latency, speed, etc. depends on the underlying application, as shown in Table 1.

Table 1 Features and requirements of various deep-learning application areas.

Application	Required latency	Required accuracy	Cost	
Military	Low	High	High	
Medical sciences	Medium	High	Medium-high	
Video surveillance	Low	Medium	Medium-high	
Agriculture	High	Low	Low	
Digit classification	Medium-high	Low-medium	Low	
Stock market	Low	High	High	

A major challenge facing deep learning researchers is the growing complexity of neural networks, which makes them unsuitable for execution on general-purpose processors. It is a fact that deep learning has traditionally been carried out on general-purpose computers. However, with time, neural networks have grown extremely large and deep. Therefore, modern neural networks cannot be efficiently trained and/or executed on a general-purpose computer (Lacey, Taylor & Areibi, 2016; Merolla et al., 2014). For efficient processing and training, specialized hardware-based neural networks are required. Since dedicated hardware platforms such as field-programmable gate arrays (FPGAs) and application-specific integrated circuits (ASICs) offer a low-power, high-speed alternative to conventional personal computers (PCs), they are becoming more popular by the day. However, such platforms come with their own set of challenges: these platforms and costly and inflexible, and their cost-efficiency is highly dependent on the underlying algorithms. Therefore, it is of utmost importance to develop algorithms and activation functions that are friendly to the hardware.

Conventional activation functions such as sigmoid, softmax, and hyperbolic tangent (TanH) yield high accuracy but are not suitable for hardware implementations. This is because they involve division and many other hardware-inefficient operations (Wuraola, Patel & Nguang, 2021). Though rectified linear unit (ReLU) (Nair & Hinton, 2010) is an extremely powerful activation function that does not require any costly elements and is the most hardware-friendly function to date, sometimes it does not produce good results. This is because it suffers from dying neurons, since it cancels out all the negative input values (Lu, 2020). If output neurons receive negative inputs only, the system will always produce zero for all the output neurons, and no sample will be correctly classified. This is why scientists have come up with functions that are not only accurate but are friendly to hardware platforms. Such activation functions do not involve any costly functions such as exponentials or long divisions (Wuraola, Patel & Nguang, 2021). Two of these functions are Sqish and square logistic sigmoid (Log_SQNL). Both these functions do not require any storage element or division operation. This is the reason why we adopt these functions for neuronal implementation in the proposed system. The Sqish and LogSQNL functions are shown in Figs. 1A and 1B, respectively.

Figure 1 Sqish and Log SQNL functions along with their derivatives.

(A) Sqish function and its derivative. (B) Log SQNL function and its derivative.

In this article we present a system based on Sqish and square logistic sigmoid (Log_SQNL) functions (Wuraola, Patel & Nguang, 2021) for breast cancer classification. The system is described in Verilog language at the register-transfer level (RTL), is implemented on a low-end Virtex 6 FPGA, and can classify a given sample, with 98.23% accuracy, into one of two classes: benign and malignant. Since most of the non-image datasets regarding disease recognition have two classes and less than 30 features, the proposed system can be used for the diagnosis of almost all diseases. The system is programmable; to diagnose a different disease, all that a user has to do is reprogram the device and use a new set of weights. The proposed system consumes only 983 slice registers, 2655 slice look-up tables, 234 DSP48 elements, and 33 block random access memories (BRAMs). The system is about 5–16 times cheaper and at least four times speedier than many modern systems such as Sarić et al. (2020), Farsa et al. (2019), Shymkovych, Telenyk & Kravets (2021), Thanh et al. (2016), Ortega-Zamorano et al. (2016), Zhang et al. (2020) and Tiwari & Khare (2015). These excellent results can be attributed to the following features:

• High Degree of Parallelism: all the required operations can be completed in a single clock cycle.

• Pipelining: use of pipeline registers at appropriate places in the system to improve throughput.

• Cost-Efficient Functions: use of Sqish in hidden layers and LogSQNL at the output layer. None of these functions require costly operations such as exponentials. Both these functions can be realized in hardware using combinational MAC computers. Since FPGAs contain a lot of DSP48 elements, multiplications can be efficiently performed.

• Proper HyperParameter Tuning: Hyperparameter tuning is extremely important for high network accuracy. The proposed network has been carefully tuned using the so-called ‘grid search’ (Zheng, 2015).

The rest of this article is organized as follows. ‘Literature Review’ presents a critical review of various high-performance activation functions and inference systems. The proposed scheme along with its hardware implementation is given in detail in ‘Proposed Methodology’. The test conditions and performance metrics are mentioned in ‘Test Conditions and Performance Metrics’. The results obtained by using the proposed scheme are given in ‘Results and Discussion’; the system is also compared against other state-of-the-art systems to prove that the proposed scheme outperforms other schemes when it comes to classification accuracy, precision, recall, and hardware efficiency. The discussion is concluded in ‘Conclusion’.

Literature Review

Since high-accuracy hardware systems are in demand, various hardware-friendly algorithms and high-performance hardware ANN systems have been presented in the literature. It is to be mentioned, however, that only a few hardware-based systems for disease diagnosis are presented in the literature. Most of the algorithms concerning medical sciences, healthcare, and disease diagnosis are not intended for any hardware implementation. For example, the algorithm presented in Kilicarslan, Adem & Celik (2020) can perform cancer classification but the authors test their scheme only on software.

A recently-developed activation function is ‘swish’ (Ramachandran, Zoph & Le, 2018). According to available reports, swish is more accurate than ReLU, especially when the network is very deep. Unlike ReLU, it is universally differentiable, i.e., the function has a valid derivative at all points on the real line. Like ReLU, the swish activation function solves the gradient vanishing problem. Swish allows negative values to backpropagate to the input side, which is impossible in the case of ReLU, since ReLU completely cancels out the negative values (Nair & Hinton, 2010). However, swish is not a hardware-friendly function since it involves division and many other costly elements (Wuraola, Patel & Nguang, 2021).

In Sarić et al. (2020), the authors present an FPGA-based system that can predict two types of epileptic seizures. Moreover, the system can predict whether a seizure is present in the first place. The overall accuracy of the system is around 95.14%. The system in Shymkovych, Telenyk & Kravets (2021) implements a simple neural network that has 4-5 synapses. The system has four Gaussian neurons, which are radial basis functions (RBFs). The system has not been tested on any well-known dataset and the purpose of the system is to demonstrate the hardware efficiency of the proposed scheme.

A high-performance activation function based on exponential units is proposed in Clevert, Unterthiner & Hochreiter (2015) that obviates the need for batch normalization. Batch normalization is an extremely costly process that requires big storage elements as well as large computational elements. Therefore, ELU is a good function in that context. However, ELU suffers from the same problems that many other activation functions do: ELU is still a cost function that is not as hardware-efficient as ReLU.

Another recently-proposed activation function is ReLTanH (Wang et al., 2019). According to its developers, it has all the nice qualities possessed by hyperbolic tangent (TanH) and at the same time, it solves the problem of gradient vanishing. A big flaw in their work is that they apply the proposed function only to the diagnosis of rotating machinery faults. They do not perform any extensive tests. Moreover, they do not implement their scheme on any dedicated hardware platform, due to which it is quite hard to determine the hardware efficiency of their algorithm and functions. However, one thing that can certainly be said about their function is that the function is not friendly to the hardware because it, like TanH, requires division and other costly operations.

A hardware system for weed classification is proposed in Lammie et al. (2019). The system finds applications in agricultural robots. The system they design uses binary weights (±1). Due to this property, the system can operate with 98.83% accuracy while having small computational units and storage elements. Eyeriss is another system that relies on extensive data reuse to reduce energy consumption (Chen et al., 2016). The system uses convolutional neural networks(CNNs) along with row- and column-wise data reuse techniques. In this way, the system achieves both high accuracy and low energy consumption. In Tiwari & Khare (2015), the researchers first implemented the sigmoid function using the ‘Coordinate Rotation Digital Computer (CoRDiC)’ technique and then implemented a complete neural network having 35 synapses using such CoRDiC neurons. The minimum value of the root mean squared error (RMSE) between the CoRDiC sigmoid and the original sigmoid is 1.67 E-11.

A hardware-based radial basis function neural network (RBF-NN) capable of online learning is proposed in Thanh et al. (2016). The network has 20 synapses and uses stochastic gradient descent (SGD) for on-chip learning. To increase hardware efficiency, the exponential terms are approximated using Taylor series expansion and look-up tables. The system has been implemented on a Cyclone-IV FPGA. The forward computation component consumes 14,067 logic elements and the SGD learning algorithm component consumes 17,309 logic elements. A comprehensive comparison of various modern works is presented in Table 2. In Table 2, C & R stands for classification and regression.

Table 2 Summary of the related work.

	Neuron	Algo.	Learning platform	Implem. platform	Accuracy	Synapses	H/W efficiency	Application	
Ramachandran, Zoph & Le (2018)	Swish	BP	Software	Software	Extr. High	–	Extr. Low	C&R	
Sarić et al. (2020)	Sigmoid	BP	Software	FPGA	High	–	Mod. High	Epil. Seizure C.	
Shymkovych, Telenyk & Kravets (2021)	Gaussian	BP	Software	FPGA	High	4–5	Mod. High	Classification	
Clevert, Unterthiner & Hochreiter (2015)	ELU	BP	Software	Software	Extr. High	–	Moderate	C&R	
Wang et al. (2019)	ReLTanH	BP	Software	Software	High	–	Low	Fault Diagnosis	
Lammie et al. (2019)	Binary	BP	Software	FPGA	High	–	High	Weed Classif.	
Chen et al. (2016)	Mixed	BP	Software	ASIC	High	–	High	Classification	
Tiwari & Khare (2015)	Sigmoid	–	–	FPGA	High	35	Low	C&R	
Thanh et al. (2016)	Radial	SGD	FPGA	FPGA	–	20	Low	C&R	

Proposed Methodology

In order to understand how the proposed system works, it is extremely important to get familiarized with a few basic concepts regarding ANN operation. Therefore, we first explain the basic ANN operation and then explain the proposed ANN topology, learning scheme, and the proposed hardware system along with its constituent components.

Basic ANN operation

Network inputs

The input values are first standardized in order to make them zero-centered. The process of standardization follows Eq. (2). In Eq. (1), X represents the input vector, µrepresents the average, and σ represents the standard deviation of data samples. The process of standardization is visually represented in Fig. 2. It is important to note that standardization is sometimes referred to as ‘normalization’ in literature, though normalization is, in reality, different from standardization.

Figure 2 Standardization of input data (Stanford University, 2022).

(1) Xstandard=Xoriginal−μσ.

Accumulation and activation

These normalized/standardized inputs are multiplied by the corresponding weights and the resulting products are then summed up (accumulated). A neuron is activated or deactivated based on the value of this weighted sum.

The activation of a neuron is dictated by a so-called ‘activation function’. Some of the various popular activation functions are rectified linear unit (ReLU) (Nair & Hinton, 2010), swish (Ramachandran, Zoph & Le, 2018), exponential linear unit (ELU) (Clevert, Unterthiner & Hochreiter, 2015), among others. Every activation function has its own merits and demerits. ReLU, for example, is used to solve the ‘gradient vanishing’ problem that occurs in hidden layers during learning. However, ReLU completely cancels out the negative region, due to which functions like swish were developed. A detailed discussion on this topic can be found in Ramachandran, Zoph & Le (2018).

The output values produced by the activated neurons are then multiplied by the corresponding weights of the next layer and the process is repeated. To offset a neuron’s value for better learning, a bias term bj is added to the weighted sum.

Classification and backpropagation

At the output layer, the neuron that is activated the most corresponds to the predicted class. The prediction of an input sample corresponds to the completion of a single iteration of the forward pass.

In the backward pass, synaptic weights are modified according to an algorithm called ‘backpropagation’. The basic idea is that the magnitude of synaptic weight updates is dictated by the magnitude of output error. If a wrong prediction is made, the error (such as a mean squared error) is computed and the synaptic weights corresponding to that (wrong) neuron are decreased. At the same time, the synapses corresponding to the correct output neuron are increased. With time, the network improves itself and eventually achieves convergence. This algorithm will be explained at length in the coming sections.

Proposed network topology and actuators

In this work, we have chosen the Sqish activation function for the hidden layer, and square logistic sigmoid (LogSQNL) for the output layer. The Sqish function is morphologically similar to swish, and LogSQNL is similar in behavior to the traditional sigmoid. The beauty of these functions is that both of these can be implemented in a single cycle using arithmetic and logic units (ALUs) only. As shown in Fig. 3, the proposed system can take 30 (or less) input features, has five hidden neurons, and two outputs.

The RTL schematic diagrams of the complete system and the predictor are shown in Figs. 4 and 5, respectively. The rectangular yellow and red boxes shown in this figure represent storage elements, and white rectangular/square boxes represent computational blocks. As shown in Fig. 5, the activation values corresponding to the two output neurons are given as input to the predictor. The predictor then calculates the maximum value and the class corresponding to this maximum value is the predicted class. The top level diagram of the proposed system is shown in Fig. 6. In Fig. 6, IM represents Mth input. The multiplier units (MUs) are responsible for multiplying weights with incoming inputs and the accumulator is responsible for adding these products. The output of the multiplier-accumulator (MAC) unit is sent to the appropriate actuator for neuronal activation.

Figure 3 The proposed ANN topology.

Figure 4 RTL schematic of the complete system.

Figure 5 RTL schematic of the predictor.

Figure 6 Top-level view of the proposed hardware system.

Mathematical setup

Now we provide details of the complete mathematical setup. Here, weights are denoted by Wi and inputs are denoted by Xi. Biases are represented by bj, and the weighted sum is represented by Zj. Finally, activation values are represented by Aj. Here, the subscript j represents the postsynaptic neuron and the subscript i represents the presynaptic neuron. The following equations represent the complete mathematical process. Z1= ∑iWi⋅Xi+b1.

Since Sqish is used as the activation function for Layer 1 in the proposed scheme, A1(Z1) is given by the following equation: A1=Z1+Z1232Z1≥0Z1+Z122−2≤Z1<00Z1<−2.

The Layer 1 activation vector is then passed as input to Layer 2 in order to obtain the weighted sum Z2, as shown in the following equation. Z2= ∑iWi⋅A1+b2.

The LogSQNL neurons in the output layer are activated according to the following rule: A2=1Z2>2Z2−Z22412+120≤Z2≤2Z2+Z22412+12−2≤Z2<00Z2<−2.

The derivative of LogSQNL function is given by Eq. (2) and the dependence of the loss function on Layer 2 weight vector and bias vector is given by Eqs. (3) and (4). The derivative of the Sqish function is given by Eq. (5) and the dependence of the loss function on Layer 1 weight vector and bias vector is given by Eqs. (6) and (7). (2) ∂A2∂Z2=2−Z240≤Z2≥22+Z24−2≤Z2<00otherwise

(3) ∂L∂W2=A2−y⋅A1⋅2−Z240≤Z2≥2A2−y⋅A1⋅2+Z24−2≤Z2<00otherwise

(4) ∂L∂b2=A2−y⋅2−x40≤Z2≥2A2−y⋅2+x4−2≤Z2<00otherwise

(5) ∂A1∂Z1=1+Z116Z1≥01+Z1−2≤Z2<00otherwise

(6) ∂L∂W1=A2−y⋅A1⋅2−Z2⋅W2⋅16+Z1⋅X1640≤Z1≤2;0≤Z2≤2A2−y⋅A1⋅2+Z2⋅W2⋅1+Z1⋅X14−2≤Z1<0;−2≤Z2<00otherwise

(7) ∂L∂b1=A2−y⋅A1⋅2−Z2⋅W2⋅16+Z1640≤Z1≤2;0≤Z2≤2A2−y⋅A1⋅2+Z2⋅W2⋅1+Z14−2≤Z1<0;−2≤Z2<00otherwise.

Proposed hardware system

As mentioned before, there are two layers in the proposed system: Layer 1 and Layer 2. Both these layers have their memories to store weights. It is pertinent to mention that all the weights are stored in the chip. The total number of weights in the system is 160 since there are 160 synapses. The on-chip weight memory consumes 1.1 kilobits. The required weights that are fetched from the corresponding memory are then multiplied by the respective layer inputs using the multiplier units (MUs) shown in Fig. 6. The multiplication is carried out using the built-in DSP48 elements. The resulting products are then summed up using an accumulator that contains an array of adders. This process is carried out for all the neurons in a layer. In the end, N weighted sums are obtained, where N represents the number of neurons in a layer. These N weighted sums are passed to their respective actuators for neuronal activation. The actuator then passes these calculated values onto the next layer and the same process is repeated. The complete structure of the proposed hardware system is shown in Fig. 6.

As mentioned earlier, Sqish neurons are used in Layer 1 and LogSQNL neurons are used in Layer 2. The structure of a Sqish neuron is shown in Fig. 7 and that of a LogSQNL neuron is shown in Fig. 8. A Sqish neuron can be implemented using two multiplexers (MUXes), two adders, two shifters and two multipliers. A LogSQNL neuron, on the other hand, consumes more resources than Sqish. A LogSQNL neuron can be implemented in hardware using three MUXes, four shifters, four adders, and two multipliers. However, since there are only two outputs in the proposed system for binary classification (one-hot encoding), the implementation of LogSQNL neurons is not a big deal.

Figure 7 Internal structure of the Sqish (hidden) layer.

Figure 8 Structure of the output LogSQNL array.

Interestingly, the distribution of data in the dataset under consideration, i.e., Wisconsin Breast Cancer (WBC) (University of California, 2022) is highly non-uniform. The standard deviation of the data is extremely large. The final input values obtained after standardization require 11 bits, where four bits are reserved for the integer part and seven bits are reserved for the mantissa (fractional part). As per our observations and calculations, the classification accuracy is more sensitive to the fractional part than the integer part. Therefore, more bits are reserved for the mantissa.

Test Conditions and Performance Metrics

In this section, we mention the test conditions under which the evaluation and comparisons are carried out. We also mention the philosophy behind the performance metrics used for evaluation.

Test conditions

Since the system has been developed for cancer diagnosis, the dataset used for experimentation is Wisconsin Breast Cancer (WBC) (University of California, 2022). This dataset has 30 features, 569 samples and two classes (benign and malignant). As per rules, about 80% samples have been used for training and 20% have been used for evaluation. To achieve class balance, some samples are picked up from the ‘benign’ class and others are picked up from ‘malignant’ class. We use Python for evaluation of the proposed scheme. The hardware is described in Verilog language at the register transfer level (RTL). The learning rate is kept equal to 13. The momentum is equal to 0.9. The data is processed in batches to achieve high accuracy; the batch size used in the proposed system is 100. The network is trained for 4300 epochs. All these hyperparameter values have been found through empirical tuning using the so-called ‘grid search’ (Zheng, 2015). The specifications of the platform on which all the tests are carried out are given in Table 3.

Table 3 Specifications of the platform used for performance evaluation.

Processor	Intel Core i7-5500 (4 CPUs)	
Memory	8.00 GB	
Operating System	Windows 8.1	
System Type	x64 (64-bit OS)	

Performance metrics

The metrics used for the evaluation of the proposed scheme are classification accuracy, precision, recall, hardware implementation cost, and system throughput. The classification accuracy is simply defined as the number of correctly-classified samples out of the total number of samples. In the context of disease diagnosis, accuracy is not a good measure of system performance. Therefore, we use precision and recall in order to properly quantify performance. The precision and recall are defined in Eqs. (8) and (9) respectively. Since these metrics are very common, we believe there is no need to discuss them in detail here. In Eqs. (8) and (9), TP stands for ‘true positive’, TN stands for ‘true negative’, FN stands for ‘false negative’, and FP stands for ‘false positive’. (8) Precision=TPTP+FP

(9) Recall=TPTP+FN.

To evaluate hardware efficiency, we use two metrics: the number of resources (number of slice registers, number of slice look-up tables, number of block memories, and DSP elements) consumed by the system, and system throughput. The throughput is defined in two ways: the number of multiply-and-accumulate (MAC) operations that can be performed in a second, and the number of input samples that can be processed by the system in a second.

Results and Discussion

Here, the proposed system is compared with other state-of-the-art systems such as Sarić et al. (2020); Farsa et al. (2019); Shymkovych, Telenyk & Kravets (2021); Thanh et al. (2016); Ortega-Zamorano et al. (2016); Zhang et al. (2020); Tiwari & Khare (2015) in terms of classification accuracy, throughput, and implementation cost. We demonstrate how the proposed scheme is better than other traditional as well as contemporary schemes, especially for disease diagnosis.

Classification accuracy, precision, and recall

As per obtained results, the system can predict the type of cancer with 98.23% accuracy. Moreover, the average precision of the proposed system is 97.5% and recall is around 98.5%. The classification report and the confusion matrix for the proposed system are given in Tables 4 and 5, respectively. Moreover, the proposed system is compared with many other state-of-the-art systems in terms of classification accuracy in Table 6. The classification accuracy as a function of epochs is presented in Fig. 9A, and the confusion matrix is visually shown in Fig. 9B.

Table 4 Classification report: the proposed system.

	Precision	Recall	
0 (benign)	0.95	1.00	
1 (malignant)	1.00	0.97	
	0.975	0.985	

Table 5 Confusion matrix: the proposed system.

TP = 38	FP = 0	
FN = 2	TN = 73	

Table 6 Classification accuracy comparisons.

	Wuraola, Patel & Nguang (2021)	Aljarah, Faris & Mirjalili (2018)	Sarić et al. (2020)	Zhang et al. (2020)	Farsa et al. (2019)	Ortega-Zamorano et al. (2016)	Proposed	
Features	784	5	8	4	25	4–35	30	
Classes	10	3	2	3	2	2–6	2	
Synapses	102k	96	153	144	130	≥84	160	
Samples	70k	822	699	1,000	8	<1,000	569	
Accuracy (%)	96.71	95.14	98.32	96	73–89	88.26	≈98.23	

Figure 9 Accuracy, precision, and recall yielded by the proposed system.

(A) Accuracy as a function of Epochs. (B) Confusion matrix—the proposed Scheme.

Implementation cost and throughput comparisons

The use of Sqish and Log_SQNL (Wuraola, Patel & Nguang, 2021) allows the processing of one sample in one clock cycle. In a single cycle, the system can perform all MAC operations and can activate all the neurons without using any divider or storage element. The multiplication operations can be performed using DSP48 multipliers that are abundantly available in an FPGA.

There are 160 synapses in the proposed neural system that operates at 63.487 MHz. The number of synaptic multiplications and additions to be performed are 160 and 153 respectively. Therefore, the system can perform 20 giga-operations in a second (GOPS). Since the system can classify one cancerous sample in one cycle (≈15.75 ns), the system can classify about 63.5 × 106 (63.5 million) samples in a second. Since a sample contains 30 inputs, about 1.91 × 109 1-input samples can be classified by the proposed system in one second. The system is compared with other state-of-the-art systems in terms of implementation cost and throughput in Tables 7 and 8 respectively.

Table 7 FPGA implementation cost comparisons.

System	Acc.	Synapses	S. Regs.	S. LuTs	Max. Freq.	Mults.	Platform	Learning	
Farsa et al. (2019)	89%	130	1023	11,339	189 MHz	–	Virtex 6	Offline	
Ortega-Zamorano et al. (2016)	88.3%	84	6766	13,062	Variable	12	Virtex 5	Online	
Sarić et al. (2020)	95.14%	96	114	12,960	50 MHz	116	Cyclone IV	Offline	
Tiwari & Khare (2015)	–	35	1898	3,124	–	154	Virtex 5	Offline	
Shymkovych, Telenyk & Kravets (2021)	–	5	790	1195	10 MHz	14	Spartan 3	Offline	
Prop.	≈98.23%	160	983	2655	63.49 MHz	234	Virtex 6	Offline	

Table 8 Throughput (TP) comparisons.

System	Synapses	Sample size	NTP	
Farsa et al. (2019)	130	25	4.73 × 109	
Sarić et al. (2020)	96	5	0.25 × 109	
Shymkovych, Telenyk & Kravets (2021)	5	4	0.04 × 109	
Proposed	160	30	1.91 × 109	

Conclusion

This article presents a high-throughput, hardware-efficient training scheme that uses Sqish neurons in the hidden layer and sigmoid-like LogSQNL neurons in the output layer. Since these functions do not require multiple cycles to process, the proposed system—based on these functions—does not consume a lot of hardware resources and yields high throughput. With only 160 synapses, the system can classify a cancerous sample into one of the two classes: benign and malignant. The proposed hardware system requires only 1.1 kilobits of on-chip memory, and can process about 1.91 × 109 1-input samples in a second. In just one second, the system can process 63.5 million cancer samples, and can perform 20 × 109 MAC operations. The system is about 5–16 times cheaper and at least four times speedier than most state-of-the-art hardware solutions designed for similar problems. Moreover, the system is way more accurate than most contemporary systems. An important item worth mentioning here is that to improve accuracy even by 1%, a lot of extra hardware resources are required. Therefore, the improvement in accuracy obtained by using the proposed scheme must not be undermined. Though the proposed system is specifically designed for cancer classification, the system can perform binary classification on any data sample that has 30 features or less. This is because the proposed system uses reconfigurable memory that can be programmed using an external computer. In future, convolutional neural networks can be applied to high-resolution mammograms (and/or ultrasound images) for diagnosing COVID-19, cancer, and other ailments.

Additional Information and Declarations

Competing Interests

Author Contributions

Data Availability

Muhammad Aleem is an Academic Editor for PeerJ.

Ali Siddique analyzed the data, performed the computation work, prepared figures and/or tables, and approved the final draft.

Muhammad Azhar Iqbal performed the experiments, analyzed the data, performed the computation work, prepared figures and/or tables, and approved the final draft.

Muhammad Aleem performed the computation work, prepared figures and/or tables, authored or reviewed drafts of the article, and approved the final draft.

Jerry Chun-Wei Lin conceived and designed the experiments, prepared figures and/or tables, authored or reviewed drafts of the article, and approved the final draft.

The following information was supplied regarding data availability:

(a) https://scikit-learn.org/stable/modules/generated/sklearn.datasets.load_digits.html.

(b) https://scikit-learn.org/stable/modules/generated/sklearn.datasets.load_breast_cancer.html#sklearn.datasets.load_breast_cancer.

(c) https://scikit-learn.org/stable/modules/generated/sklearn.datasets.fetch_openml.html.

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
