# Peer review of "A high-performance, hardware-based deep learning system for disease diagnosis"

_PeerJ Computer Science, doi:10.7717/peerj-cs.1034_

## Round 0.1 · original submission · Major Revisions

Please see all reviewers comments and revise the paper accordingly.

Reviewer 1 ·

Basic reporting

The paper proposes a novel methodology to diagnose cancer. The abstract should be verified in order to remove the grammatical errors.
Which type of cancer and from which organ has to be explained in detail.
The abstract states that the proposed system can also detect diseases from the heart. This sentence can be removed or the evidence for this sentence could be included.
Table 1 must be shown in the introduction section as it is not related to the literature review. Rather, a summary of the literature can be included as a separate table in this section.
The proposed solution can be written as Proposed Methodology
More details can be included in the Basic ANN Operation as it plays a major role in this paper.

Experimental design

It will be well if the experiments were done for the proposed system.
A table explaining the values of TP, FP, and TN for the input data should be included.

Validity of the findings

Table 3: Comparison of Various Neural Networks in terms of Classification Accuracy has to be rewritten so that it can be easily understandable to the readers.

Reviewer 2 ·

Basic reporting

More points have to be included based on the complete methodology and results
The sentence “Moreover, the system is reconfigurable and can be programmed to classify any sample into one of two classes” has to be justified as the classification is nowhere shown throughout the manuscript.
The sentence “The system can predict about 63.5 million cancer samples in a second” has to be justified as the results were not shown in the results and discussion section. Rather, it is shown only in the abstract and also in the conclusion.

Experimental design

An experiment has to be done in order to depict the prediction of 63.5 million cancer samples in a second. Also, a tabulation has to be framed to analyze the prediction accuracy.

Validity of the findings

It will be well if executed results of image samples were shown in the results and discussion section so that the proposed methodology seems to be executed.

Reviewer 3 ·

Basic reporting

The paper proposed a hardware implementation for the cancer diagnosis problem. However, the contribution is not clear in the hardware methodologies that are introduced

1- The proposed architecture is very weak and has no enough contribution
as in Figure 3: Proposed ANN Topology
2- The proposed system is not clear from Figure 6: Internal Structure of the Proposed Hardware System.
3- Figures 4,5, and 7 are not clear.

Experimental design

1- The experiments should include several cancer datasets and evaluate the performance between them
2- The splitting technique is very weak, I think you should use 10 folds cross-validation to get fair comparison.
3- The training parameters needs to be illustrated why u choose these parameters
The learning rate is kept equal to 1/3 . The momentum is equal to 0.9. The data is processed
in batches in order to achieve high accuracy; the batch size used in the proposed system is 100.

Validity of the findings

The paper did not contribute enough to be published in the journal and needs to high modification level

Reviewer 4 ·

Basic reporting

Authors have proposed a hardware-based deep learning system for cancer diagnosis. The research problem selected by author(s) is timely and important to be addressed. The overall structure and organization of the paper are satisfactory, and the paper qualifies for an above-average up-to-date bibliography. However, there are a few issues that are required to be addressed by the authors.


1) The main problem with this paper is coherency. In many paragraphs, sentences are not written coherently.
2) Why the title is specific to Cancer Diseases? Authors have also claimed the ability of the proposed scheme to predict the presence of other diseases. So, the title should be a hardware-based deep learning system for disease diagnosis.
3) Authors are encouraged to improve the abstract by focusing only on the most significant details that are unique to their proposal. Authors have simply claimed in the abstract that “In this context, we propose a hardware-based neural network that can predict the presence of cancer in humans with 98.23% accuracy.” But they didn’t mention the features of their system that help to gain an accuracy of 98.23%. Authors are advised to write their significant contributions clearly. Moreover, in the abstract, it has been mentioned that the proposed system is about 5 to 16 times cheaper and at least four times speedier than many other contemporary systems? What is meant by many? Not clear.
4) Authors have several times used the term hardware friendly but have not described it. For example, in the abstract and other sections, it is stated “this is why scientists have come up with functions that are not only accurate but are friendly to hardware platforms”. What is meant by friendly to hardware platforms? Any reference?
5) “Conventional activation functions such as sigmoid and hyperbolic tangent (TanH) yield high accuracy but are not suitable for hardware implementations. This is because they involve division and many other hardware-inefficient operations.” Authors have mentioned this fact, but without reference(s).

Experimental design

6) It seems there is no need for this sentence “To know more about
efficient implementation of neural networks on edge devices, the reader is referred to [11].” Is the implementation of neural networks on edge devices is related to the authors’ work?
7) Authors have claimed “this is the reason why we adopt these functions for implementation in the proposed system. Adopted these functions for implementation of what? in the proposed system. It is not clear here.
8) A summary or comparison table of proposed techniques discussed in the literature review is not available.
9) Details in paragraphs 2, 3, and 4 of Section 1 are not coherent. It is highly recommended to re-write these paragraphs to highlight your contribution.

Validity of the findings

10) The required level of accuracy, speed, etc. depends on the underlying application, as shown in Table 1. Delay is not mentioned and how this line is related to the other lines of this paragraph. Again, authors are advised to be coherent in writing paragraphs.
11) Section 3 proposed solution starts with the working of the proposed system but authors must add text about the proposed techniques here. What is the proposed system, components, and working?
12) “Data Preprocessing” heading is not required and the text under this heading should be adjusted in section 3.1.
13) Figure 4 is not clear.
14) Figure 6 should be discussed in section 3.2.
15) The proposed algorithm is compared with other state-of-the-art algorithms in terms of classification accuracy, throughput, and implementation cost. Which state-of the art algorithms? Names?
16) The proposed system consumes only 983 slice registers, 2655 slice look-up tables, 234 DSP48 elements, and 33 block random access memories (BRAMs). But why? It has not been mentioned.
17) It can be seen from Table 4 and Table 5 that the proposed system is about 5-16 times cheaper and at least four times speedier than most modern systems. What modern systems? Mention features of your proposed solutions that make these results better.
18) Future work or extension is not clear. Just one line has been mentioned, “In future, more complex datasets can be chosen for better diagnosis.” This is not enough to say what is meant by complex datasets?

---

## Round 0.2 · accepted · Accept

All of the previous comments have been addressed.

Reviewer 1 ·

Basic reporting

All requirements have been revised satisfactorily.

Experimental design

All requirements have been revised satisfactorily.

Validity of the findings

All requirements have been revised satisfactorily.

Reviewer 2 ·

Basic reporting

All comments have been addressed

Experimental design

All comments have been addressed

Validity of the findings

All comments have been addressed

Additional comments

All comments have been addressed

Reviewer 4 ·

Basic reporting

Authors revised the paper, so all comments were met

Experimental design

Authors revised the paper, so all comments were met

Validity of the findings

Authors revised the paper, so all comments were met